# Total Phenolic Content and Antioxidant Activity of In Vitro Digested Hemp-Based Products

**DOI:** 10.3390/foods12030601

**Published:** 2023-02-01

**Authors:** Davide Lanzoni, Eva Skřivanová, Raffaella Rebucci, Antonio Crotti, Antonella Baldi, Luca Marchetti, Carlotta Giromini

**Affiliations:** 1Department of Veterinary and Animal Sciences (DIVAS), Università degli Studi di Milano, Via dell’Università 6, 29600 Lodi, Italy; 2Department of Microbiology, Nutrition and Dietetics, Faculty of Agrobiology, Food and Natural Resources, Czech University of Life Sciences Prague, Kamycka 129, 165 00 Prague, Czech Republic; 3Department of Nutritional Physiology and Animal Product Quality, Institute of Animal Sciences, Pratelstvi 815, 104 00 Prague, Czech Republic; 4CRC, Innovation for Well-Being and Environment, Università degli Studi di Milano, 20122 Milano, Italy

**Keywords:** antioxidant, Folin–Ciocalteu, function profile, hemp, in vitro digestion, phenolic compounds

## Abstract

The growth of the world population has prompted research to investigate new food/feed alternatives. Hemp-based products can be considered excellent candidates. Hemp (*Cannabis sativa* L.) is an environmentally sustainable plant widespread worldwide. Following the reintroduction of its cultivation, hemp is attracting interest, especially in the food/feed industry. To date, scientific research has mainly focused on its nutritional aspect. Therefore, the aim of the work was also to investigate the functional profile (total phenolic content (TPC) and antioxidant activity (Ferric- reducing antioxidant power (FRAP) and 2′-azinobis-(3-ethylbenzothiazoline-6-sulfonic acid (ABTS)) of hemp-based products (hempseeds (HSs), flowers, and HS protein extract), following methanol extraction and in vitro digestion, to study the behaviour of the molecules involved. The results show an interesting nutritional value, even when compared to matrices used in the food/feed industry, such as soy and flaxseeds. The functional profile revealed a very interesting TPC following methanol extraction for HSs, flowers, and HS protein extract, respectively, (550.3 ± 28.27; 2982.8 ± 167.78; and 568.9 ± 34.18 mg Tannic Acid Equivalent (TAE)/100 g). This trend was also confirmed for FRAP (50.9 ± 4.30; 123.6 ± 8.08; and 29.73 ± 1.32 mg Ascorbic Acid Equivalent (AAE)/100 g), recording values similar/higher than soy protein extract and flaxseeds (17.4 ± 1.55; and 10.4 ± 0.44 mg AAE/100 g). The results were also maintained following physiological digestion. These results, although promising, need further investigation, confirming what has been observed with different antioxidant activity assays and identifying individual molecules involved in functional pathways. This information will be necessary to gain a better understanding of the functional characteristics of these matrices for use in food/feed formulations.

## 1. Introduction

The steady growth of the world population will reach 9–11 billion by 2050. At the same time, there will be a dramatic growth in demand for food, reaching 50% by 2030 and doubling by 2050 at which point it will be difficult to supply the demand without negatively impacting the health of the environment [1,2,3]. The main products required will be those of animal origin, in particular meat and dairy products. According to statistics from the Food and Agriculture Organization of the United Nations (FAO), the world demand for meat will reach 455 million tonnes by 2050, an increase of 76% compared to 2005 [2]. As a result, there has been a substantial increase in the need for feed for the livestock sector, opening the feed/food competition debate. Today, around 800 million tonnes of cereal are used within the feed sector, a value that will exceed 1.1 billion tonnes by 2050. Expansion of the monogastric sector will lead to an increase in demand for maize and coarse grains, accounting for almost half of the cereals produced in 2050. In 2000 alone, pigs and poultry consumed about 78% of feed grains. More precisely, in 2013, the monogastric sector consumed 155 million tonnes of proteins, and by 2030, an additional 52 million will be needed to satisfy the sector’s demand. Most of these are in direct competition with human nutrition [4]. The necessity has therefore arisen to satisfy the nutritional requirements of a rapidly increasing population through sustainable and environmentally friendly processes. This goal is strongly pursued by the European Union (EU), which, first with the One Health approach and later with the Green Deal (1 December 2019), aims to recognize that human, animal, and environmental health are interconnected, promoting efficient use of resources towards a clean and circular economy [5,6]. For this reason, to ensure food safety and environmental protection, research in recent years has been largely focused on finding new food/feed alternatives that meet the principles proposed by the EU.

Among the possible alternatives, hemp (*Cannabis sativa* L.) is attracting great interest. Hemp, originated in Central Asia and considered to be one of the oldest crops, is a dicotyledonous annual, herbaceous, and angiosperm plant widespread throughout the world due to its easy adaptability [7,8]. It is a low environmental impact crop, traditionally cultivated for fibre production. Despite its rapid expansion, the hemp trade declined sharply between the 1940s and the late 1960s, disappearing from the market, as a result of both the introduction of synthetic fibres and, above all, the entry into force of the Protocol on the Single Convention on Narcotic Drugs proposed by the United Nations, signed in New York on 30 March 1961 and subsequently the Protocol of Amendment adopted in Geneva on 25 March 1972. This convention put an end to the cultivation of hemp in more 183 countries [9]. However, at the start of the 21st century, interest in hemp was relaunched thanks to the introduction of EU Regulation 1251/1999/EC of 17/05/1999, which instituted a support system for all producers of particular types of arable crops, among them industrial hemp. However, it was only with Regulation (EU) No 1307/2013 that hemp cultivation was also included among those qualifying for Common Agricultural Policy (CAP) payments, with the only condition being that the seeds used for cultivation be of varieties registered in the European catalogue with a Δ9-tetrahydrocannabinol (THC) content (i.e., the plant’s recognised psychoactive compound) of less than 0.2% [9,10]. The introduction of these laws allowed hemp to reach the market, finding use in a wide range of sectors, from construction and energy to cosmetics and pharmaceuticals. More precisely, to date, the hemp market comprises more than 25,000 products [11]. In recent decades, interest in hemp and hemp-based products has increased, also involving the food and feed sector, as reported by EFSA (2011) [10].

In particular, the food industry has investigated the use of hempseeds (HSs) to obtain oil, protein, and dietary fibre, facilitating their incorporation into food products, such as yoghurt and baked goods [11]. In addition, Guang and Wenwei (2010) [12] used HS flours to produce functional food that aid disease prevention by increasing high-density lipoprotein levels and stabilising the level of other triglycerides and lipoproteins.

As shown, the interest is in the seeds of this plant, previously considered as a waste product in fibre production, but characterised by high nutritional and functional profiles [13], prompting researchers to deepen their potential beneficial effects in human and animal diets. To date, scientific research has focused more on the nutritional characterisation. The functional aspect, however, is still little explored. For this reason, the aim of the study was to characterise not only the nutritional, but above all, the functional profile of hemp-based products, in particular total phenolic content (TPC) and antioxidant activity, following extraction with pure methanol and in vitro digestion, to study the behaviour of the molecules involved during different phases of the digestive process, thus enabling new information on the functional aspect of these matrices to be acquired.

More specifically, in our study, the characterisation of the nutritional profile involved the analysis of the protein, lipid, fibre, and ash contents, as well as the determination of the digestibility level. The functional profile characterisation, instead, entailed quantification of the TPC and antioxidant capacity.

## 2. Materials and Methods

### 2.1. Materials

Hempseeds (*C. sativa* L, variety Futura) were purchased from a Czech company (Chrastice, Czech Republic). Specifically, the hemp sowing took place in April/May 2021, and the harvest was carried out at 70% maturity. After collection, the seeds were dried at 40 °C to a final moisture content of 7% and stored at 10–12 °C in the dark. Cannabinoid analysis was performed at the Institute of Animal Science (Czech Republic) showing THC and cannabidiol (CBD) contents of 0 µg/g and 30 µg/g, respectively. HS samples, before each analysis, were stored at 4 °C in the dark.

The hemp flowers (*C. sativa* L, variety Carmagnola) were provided by a local company (CN, Italy). Specifically, they were collected from plants sown from May to October 2021 and harvested in October 2021. Manual harvesting with the removal of the cut plants was followed by slow drying in a closed, ventilated environment, without direct light, to a humidity between 12 and 14 °C. Subsequently, the flowers were analysed for THC (<0.2%) and CBD (7.24%) contents. The samples, before each analysis, were stored at room temperature (RT) in the dark.

HS protein extract, soy protein extract, and flaxseeds (*Linum usitatissimum* L.) were provided by a commercial supplier.

### 2.2. Chemical Analysis

The chemical analysis of the samples (HSs, flowers, HS protein extract, soy protein extract, and flaxseeds) was performed according to official methods [14,15], and the fibrous fraction was analytically measured as reported by Van Soest et al. (1991) [16].

### 2.3. Methanol Extraction

For methanol extraction, each sample (HSs, flowers, HS protein extract, soy protein extract, and flaxseeds) was ground and weighed (5 ± 0.5 g) and mixed with 30 mL methanol (100%) for 48 h at RT in the dark. Subsequently, each sample was filtered with filter paper (Whatman 54, Florham Park, NJ) in accordance with Castrica et al. (2018) [17] and Attard (2013) [18].

### 2.4. Total Phenolic Content and Antioxidant Activity

Subsequently, the chemical extracts were analysed for TPC and antioxidant activities (ABTS and FRAP).

#### 2.4.1. Total Phenolic Content

For the total quantification of phenols, the protocol of Attard (2013) [18] was followed. Tannic acid, methanol, Folin–Ciocalteu (FC) reagent, and sodium carbonate were purchased from Sigma Chemical Co. (St. Louis, MO, USA). Tannic acid was prepared in seven 1:2 dilutions, from 960 µg/mL down to 0 µg/mL. The FC reagent was diluted 1:10 with distilled water, while sodium carbonate was prepared as a 1 M solution. Then, 100 µL of each sample was added to 500 µL of FC and 400 µL of sodium carbonate and incubated in the dark at RTfor 20 min. At the end of incubation period, samples were read at 630 nm. Appropriate solvent blanks were run in each assay. TPC was expressed in terms of tannic acid equivalent (mg TAE/100 g).

#### 2.4.2. ABTS Assay

ABTS 2′-azinobis-(3-ethylbenzothiazoline-6-sulfonic acid) assay was performed following the protocol of Re et al. (1999) [19]. In particular, 2.5 mM Trolox (6-hydroxy-2,5,7,8-tetramethychroman-2-carboxylic acid; Sigma Chemical Co.) (St. Louis, MO, USA) was used as antioxidant standard. Fresh working standards were prepared daily on dilution with ethanol. ABTS, provided by Sigma Chemical Co. (St. Louis, MO, USA), was dissolved in distilled water to a 7 mM concentration. ABTS radical cation (ABTS^•+^) was produced by reacting ABTS stock solution with 2.45 mM potassium persulfate (final concentration) and allowing the mixture to stand in the dark at room temperature for 12–16 h before use. For the study of antioxidant activity, the ABTS^•+^ solution was diluted with ethanol to reach an absorbance value of 0.70 (±0.02) at 734 nm. Then, 20 µL of sample was added to 2.0 mL of diluted ABTS^•+^ solution (A_734nm_ = 0.700 ± 0.020), incubated for 6 min at RT in the dark, and read to 734 nm. Appropriate solvent blanks were run in each assay [20]. Values were expressed in terms of Trolox equivalent (mg TE/100 g).

#### 2.4.3. Ferric-Reducing Antioxidant Power (FRAP) Assay

FRAP assay was performed following the protocol of Abdelaleem and Elbassiony (2021) [21] with minor adaptations. In particular, ascorbic acid was used as antioxidant standard and prepared in seven dilutions from 1000 µL to 0 µL. a) Acetate buffer (300 mM; pH 3.6) was prepared using 2.69 g sodium acetate trihydrate (Sigma Chemical Co. (St. Louis, MO, USA)) in 16 mL of glacial acetic acid and made the volume to 1.0 L with distilled water. b) TPTZ (2, 4, 6-tripyridyl-s- triazine), provided by Sigma Chemical Co. (St. Louis, MO, USA), was obtained dissolving 31.2 mg in 10 mL of 40 mM of HCl. c) FeCl_3_ (Sigma Chemical Co. (St. Louis, MO, USA)) was obtained dissolving 0.054 g in 10 mL of distilled water. Then, 10 µL of each sample was added to 300 µL of FRAP reagent and incubated at RT for 10 min in the dark and read at 595 nm. Appropriate solvent blanks were run in each assay. Values were expressed in terms of acid ascorbic equivalent (mg AAE/100 g).

### 2.5. In Vitro Digestion and Digestibility

In vitro digestion was performed according to Regmi et al. (2009) [20], with minor adaptations reported by Castrica et al. (2019) [17]. The protocol aims to simulate the three phases of digestion. At the end of each digestive phase (oral, gastric, ½ intestinal, and intestinal), aliquots (1 mL) corresponding to the soluble fraction were taken and frozen at −80 °C and used to monitor the TPC and antioxidant activities during in vitro digestion. In addition, at the end of digestion, an undigested fraction (UF) was also obtained. The UF was subsequently harvested in a filtration unit using a porcelain filtration funnel covered with preweighed filter paper (Whatman 54 Florham Park, NJ). The UF, together with the filter paper, was dried overnight at 65 °C. The UF was used to determine the in vitro digestibility Equation (1):Digestibility (% dry matter; DM) = (sample DM − UF DM)/sample DM × 100(1)

The described procedure was performed in triplicate (*n* = 3). Following in vitro digestion, the samples were analysed for total phenolic content and antioxidant activity (ABTS and FRAP), as described above.

### 2.6. Statistical Analysis

All data were analysed by one-way Anova followed by Tukey’s multiple comparison test, using GraphPad Prism 9 9.3.1 (GraphPad Software Inc., San Diego, CA, USA).

All data are expressed as means ± standard error of the mean (SEM) of at least three independent experiments. Values are considered statistically significant for a 95% confidence interval (*p*-value = 0.05).

## 3. Results and Discussion

### 3.1. Chemical Analysis and In Vitro Digestibility

The chemical composition and digestibility of the matrices under study are given in Table 1.

As previously reported, HSs can be considered as one of the most complete sources from a nutritional perspective due to their high nutritional characteristics [13], as shown in Table 1. In the literature, many authors show high variability in HS composition due to genotype and environmental growth factors [22,23,24]. However, as reported by Leonard et al. (2020); Farinon et al. (2020); and Callaway (2004) [13,25,26], they typically contain 94% dry matter, 20–25% protein, 25–35% lipids, 20–30% carbohydrates (mostly dietary fibre), and 5–6% ash. These values, as show in Table 1, are comparable to those obtained for flaxseeds, one of the most important oilseed crops for industrial as well as food, feed, and fibre purposes [27,28]. HSs show a good level of digestibility, higher than that of flaxseeds, showing statistically significant differences (*p* < 0.05). Zhou et al. (2020) [29] demonstrated that in vitro digestion of flaxseed polysaccharides did not lead to their degradation (no change in molecular weight), suggesting that ingested polysaccharides would reach the large intestine intact. Similar results were reported by Marambe et al. (2012) [30]. The authors observed that following the in vitro digestion process, the protein digestibility of flaxseeds was only 12.61%, values that tended to increase following the removal of the fibrous and lipid contents. In the light of the results obtained, and as reported in the literature, to improve the digestibility of these two matrices, it is preferable not to use the whole seed, but only after treatment (heat, extrusion, and dehulling) to reduce the fibre level [25] or to prefer the meal to reduce the fat content, thus making the seeds more suitable for food and feed applications.

Hemp flowers are an important component of the plant. They are currently used for ornamental purposes as well as in cosmetics and pharmaceutical. The essential oil, rich in functional properties (antioxidant and anti-inflammatory) [13] can be obtained from their processing. Although the use of flowers within the food/feed sector is not thoroughly documented in the literature, they have an interesting nutritional profile, especially in terms of protein content, suggesting their possible use within the feed sector, as confirmed by Kleinhenz et al. (2020) [31]. However, the high fibre and mineral contents adversely affect final digestibility. Furthermore, the presence of phytocannabinoids makes it necessary to study their effects on animal and/or human health. For this reason, further investigations are needed to assess their possible inclusion within the food and feed sector.

In our experiment data, the digestibility of HS protein extract showed no statistically significant differences with soy protein extract. Mamone and colleagues (2019) [32] studied how HS protein is characterised by high digestibility. The authors demonstrated how, following in vitro digestion, only a few peptides withstood the digestive process. This finding is also confirmed by Wang et al. (2008) [33]. They not only observed that HS protein was highly digestible, but that the level of digestibility was also higher than that of soy. Protein digestibility is a crucial parameter, as together with amino acid composition and bioavailability, it determines the nutritional value of a protein source [13]. HS protein is characterised by a good amino acid content, greater than or similar to that of soy, with the exception of aspartic acid, glutamic acid, and lysine. In addition, it contains all essential amino acids [13]. This suggests that HS protein can be used as a viable substitute to soy protein, the main protein source used in the feed industry due to its important nutritional value [34].

### 3.2. Total Phenolic Content and Antioxidant Activity of Methanol Extracts

In Table 2, the values for TPC and antioxidant activities (ABTS and FRAP) of methanol extracts are reported. In our study, chemical extraction was performed in methanol to ensure the best possible yield, as demonstrated by Kalinowska et al. (2022) [35].

#### 3.2.1. Total Phenolic Content of Methanol Extracts

As shown in Table 2, the TPC of HS is comparable to that of flaxseed (550.3 ± 28.27 vs. 634.0 ± 18.95), known for their high phenol content, mainly lignans, flavonols, flavanones, flavones, and phenolic amides [36]. This result is partially confirmed by Galasso et al. (2016) [37]. The authors investigated the TPC of HS and flaxseed and reported that the levels were slightly higher in the former. This difference can be explained by the influence of several biotic and abiotic factors involved in the biosynthetic pathway of phenols, among which, genotype, year of cultivation, and the interaction between these factors [38]. More precisely, HSs used in this study are characterised by both a different year of cultivation, altitude and, above all, harvesting at a different stage of maturity, most likely factors responsible for this difference. Phenolic compounds are secondary metabolites, which plants produce as a defensive weapon against biotic and abiotic stresses, among them predator attack and UV radiation. They are known for their intrinsic antioxidant effect, which means they can protect cells from oxidative damage, thus limiting degenerative diseases associated with oxidative stress [9].

HSs as reported in the literature are characterised by a high phenol content, localised more in the hull than in the kernel [39,40], as well as in oil (flavanones, flavonols, flavanols, and isoflavones) [41]. In particular, the main phenolic compounds identified in HSs are the lignans, phenols derived from the shikimic acid biosynthetic pathway, also called phenylpropionamides thanks to their particular chemical structure. They are part of two major groups, known as phenolic amides and lignanamides [38]. Lignanamides are those most commonly found within HS, including cannabisin B and N-trans-caffeoyltyramine, which are the main phenolic compounds in the hull fraction; catechin, in the cotyledonary fraction [38], as well as cannabisin A, F, I, and Q and grossamide [40,41,42].

In the literature, the characterisation of TPC related to hemp flowers is little studied. However, as shown in Table 2, they have a higher TPC than the other analysed matrices, showing statistically significant differences. As reported by Izzo et al. (2020) [43], hemp flowers can be regarded as a promising new source of phenols for nutraceutical formulations. The authors demonstrated through their characterisation that this part of the plant is distinguished by a high content of lignanamides, specifically cannabisin A, B, and C. Additional phenolic compounds include hydrozycinnamic acids (caffeic acids, chlorogenic acid, *p*-coumaric acid, and ferulic acid) and flavonoids. Of the latter, those most commonly present are flavones, in particular, cannflavin A and B, which are characterised by important functional properties and are 10–100 times more abundant than in other parts of the plant, most likely explaining the reason for such a high TPC [43].

As shown in Table 2, the TPC of the HS protein extract is highly comparable to that of the whole seed (550.3 ± 28.27 vs. 568.9 ± 34.18). This result can be attributed to the fact that the protein extract tested is not pure but has similar fibre concentrations as the whole seed, a component where most of the phenolic compounds might be located. At the same time, the protein extract of HS is comparable to that of soy. The values show no statistically significant differences, although that of soy is higher. As reported by Chatterrjee et al. (2018) [44], isoflavones, one of the main bioactive compounds in soy, are bound to the protein component, resulting in a greater value.

However, it must be considered that non-phenolic compounds, such as some vitamins and their derivatives (L-ascorbic acid, folic acid, folinic acid, retinoic acid, and thiamine), nucleotide bases (e.g., guanine), amino acids (tyrosine, tryptophan, and cysteine), and simple inorganic ions (Fe^+2^, Mn^+2^, I^−^, and SO_3_^−2^), react as well as with the Folin–Ciocalteu reagent. Nevertheless, the assay is commonly applied for the study of the TPC in food products [35].

#### 3.2.2. Antioxidant Activity of Methanol Extracts

The antioxidant activity of phenolic acids is related to the quantity, number, and position of hydroxyl groups in the molecule. Phenolic compound can act in many ways: (I) chelating metals, such as iron and copper; (II) breaking the chain of reactions triggered by free radical; and (III) slowing down or accelerating enzyme activity [45]. Given the different modes of action of phenolic compounds, the antioxidant activity was assayed by two different methods: (I) ABTS, allows the quantification of free radical scavenging capacity and (II) FRAP, allows the quantification of compounds capable of reducing the complex of ferric ions (Fe^+3^) ligand to ferrous complex (Fe^+2^).

As shown in Table 2, HSs are characterised by good antioxidant activity, highly comparable to that of flaxseeds. No statistically significant difference was observed with the ABTS method, unlike that reported for FRAP. However, as can be seen, HSs show greater scavenging activity than metal chelation. HSs, in fact, are characterised by the presence of cannabisin B, s N-trans-caffeoyltyramine, and high concentrations of tocopherol, compounds capable of disrupting free radical chain reactions by capturing them [46,47]. However, as suggested by Chen et al. (2012) [46], the antioxidant activity is not so much related to the type of molecule present, but to the concentrations of the individual compounds within the matrix analysed.

As reported in Table 2, hemp flowers show a greater antioxidant activity than all other matrices studied, showing statistically significant differences. Again, the scavenging activity of the antioxidant molecules is higher. This result is not only due to the presence of phenolic compounds, but also to CBD within the flowers, the only component of the plant to have it. CBD, in fact, is characterised by multiple properties, including anti-inflammatory and antioxidant properties. In particular, it is able to inhibit oxidant molecules by interrupting the chain reactions triggered by free radicals, capturing them or transforming them into less active forms [48]. It can also promote a direct reduction in oxidant levels as well as modify the redox balance by changing the level and activity of antioxidant molecules [49,50].

The antioxidant activity of HS protein extract appears to be comparable to that of soy. In the literature, the functional activity of these protein matrices is not very extensive. However, as shown by Farinon et al. (2020) [13] and Chatterrjee et al. (2018) [44], they acquire high antioxidant value only, and following the process of hydrolysis by digestive enzymes are capable of fragmenting the protein structure by releasing bioactive peptides.

### 3.3. Total Phenolic Content and Antioxidant Activity of In Vitro Digested Samples

A further step in the study was the evaluation of TPC trends and antioxidant activity during the different phases of in vitro digestion (end of oral phase, end of gastric phase, ½ intestinal phase, and end of intestinal phase) to assess the bioaccessibility of these compounds.

#### 3.3.1. Total Phenolic Content of In Vitro Digested Samples

Figure 1 shows the TPC trend of all analysed matrices.

The nature of phenolic components is important in assessing the antioxidant activity of a sample [51]. As shown in Figure 1, the TPC within the analysed matrices shows the same trend, with higher values for hemp flowers, most probably due to the ready bioavailability of phenolic compounds. HS shows a higher content than flaxseed, while the two protein matrices are highly comparable.

These results are confirmed in other studies in the literature conducted on plant matrices (bamboo leaves, Butia, and Carob fruits). At the end of the oral phase, there is an initial and partial degradation of the phenolic compounds [52,53,54]. Furthermore, as reported by Leonard et al. (2020) [25], there is a strong interaction between fibre and phenols. In fact, in addition to being able to trap them, it reacts chemically through hydrophobic interactions, hydrogen bonding (oxygen atoms from polysaccharide’s glycosidic chains and hydroxyl groups from phenolic compounds), and covalent bonds, thus reducing the bioavailability of phenolic compounds within the analysed matrices. However, as suggested by Ginsburg et al. (2012), saliva could play a key role in the solubilisation of phenolic compounds by substantially increasing their availability [55]. At the end of the gastric phase, an increase in TPC can be observed in all analysed matrices. This increase is the result of a low pH value (2 ± 0.05) that can promote the release of phenols following the breaking of bonds within the matrix including polysaccharides and proteins [53]. Subsequently, in the middle of the intestinal phase, the TPC trend decreases. This result, also confirmed by Friedman and Jurgens (2000) [56], is a direct consequence of the instability of phenols at high pH values (6.8 ± 0.05). At the end of the intestinal phase, there is a slight increase in TPC, as the phenolic compounds are, most likely, transformed into other structural forms, formed as a result of the cleavage of specific bonds in their structure [57]. These molecules, detected by the assay, thus result in an enhanced concentration of TPC. In conclusion, TPC during the in vitro digestion process is thus influenced by the nutritional components, especially fibre, but above all by the physiological conditions of the digestion process. Concerning the individual matrices, it can be observed that hemp flowers also in this case showed the highest TPC. HS showed a slightly higher TPC than flaxseed, as did the HS protein extract compared to soy.

#### 3.3.2. Antioxidant Activity of In Vitro Digested Samples

For the evaluation of the effect of in vitro digestion on the antioxidant capacity of hemp-based products, the ABTS and FRAP assays of digested samples were determined and are shown in Figure 2.

As reported in Figure 2, HS shows higher (ABTS) and highly comparable (FRAP) values than flaxseed, further confirming the important functional aspect of this matrix. The same applies to HS protein extract when compared to soy protein. As previously reported, HS protein is not characterised by high bioactivity. However, the hydrolysis process in the protein structure leads to the formation and release of bioactive peptides. Bioactive peptides can be defined as isolated small fragments of protein, which provide some physiological health benefits [58]. In particular, HS peptides are characterised by high functional properties, including antioxidant, antihypertensive, antiproliferative, cholesterol-lowering, anti-inflammatory, and neuroprotective properties [59,60,61]. This shows that the peptides are encapsulated within the native protein structure and are only released during the hydrolysis process. In particular, Tang and coworkers (2006) and Wang and colleagues (2009) demonstrated that HS hydrolysates with a greater degree of hydrolysis possessed greater antioxidant activity in vitro, particularly with regard to radical scavenging and iron chelation capacity [62,63].

Hemp flowers, although they once again confirm their great antioxidant power, show a different trend from those we studied and from those reported in the literature. For this reason, it is necessary to investigate this aspect further for their possible use within the food and feed sector.

The antioxidant activity of the samples under study showed different trends for the two methods used (ABTS and FRAP). The values obtained for FRAP, the method used in other plant matrices, as reported by Koocheki et al. (2022) [64] are in line with those observed for TPC. As can be seen in Figure 2, there is a peak in antioxidant activity following the gastric phase. As explained above, this may lead to the release of phenolic compounds during the digestive process. These results are also confirmed in studies by Gonzalez et al. (2016) [65] and Ma et al. (2020) [52], where different matrices, such as Maqui berri and bamboo leaves, showed the same trend. In the middle of the intestinal phase there is a decrease in antioxidant activity, because as reported by Ma and colleagues, a large part of the phenolic compounds is degraded at this stage of the digestive process [52]. At the end of the digestive process, a slight increase in antioxidant activity can be observed, due to the release of additional compounds with high antioxidant activity, such as flavonoids, tocopherols, carotenoids, pigments, and ascorbate [57].

As far as the antioxidant activity recorded by the ABTS method is concerned, the graphs in Figure 2 all show the same trend, with the exception of the hemp flowers, which record a decrease in antioxidant activity at the end of the gastric phase. For all the other matrices, a peak is observed at the end of the intestinal phase, confirming what was previously reported. Based on the evidence of Pinacho et al. (2015) [57], phenolic compounds are sensitive to the alkaline pH of the intestinal phase and for this reason may be converted into other structural forms, either unknown or difficult to detect, that exert their antioxidant activity at the end of digestion.

These values, coupled with the functional profile, emphasise how hemp-based products can be considered as a viable alternative to flaxseeds and soy, the main crops used in the food and feed industry. In agreement with the WWF, each kilogram of chicken, pork, eggs, and meat contains 575, 263, 307, and 173 g of soy, respectively [66]. As reported in the literature, soy has a high environmental impact [67]. Hemp, which is characterised by an important role in environmental health, could mitigate this problem.

## 4. Conclusions

In the light of these reports, HSs and HS protein extract proved to be products with a high nutritional profile. Interesting results were also reported for flowers, where the high protein content suggests their possible application in the food and feed sector. However, while the nutritional aspect of hemp-based products has already been adequately described in the literature, the functional one is still little investigated. For this reason, the aim of this work was to further explore the functional characterisation of hemp-based products following extraction with pure methanol and the in vitro digestion process, obtaining new information on TPC and antioxidant activities. The reported results show that the flowers are characterised by a high functional profile and that the HSs and HS protein extract have similar or higher values than those reported for other food/feed matrices, such as soy and flaxseeds. The high nutritional aspect combined with the functional one suggests that hemp-based products can be considered as viable alternatives for the food and feed industry. However, these results, although promising, need further investigation, confirming what has been observed with different assays of antioxidant activity and identifying the individual molecules involved in the functional pathway. This information will be necessary to gain a better understanding of the functional characteristics of these matrices for use in food/feed formulations.

## Figures and Tables

**Figure 1 foods-12-00601-f001:**
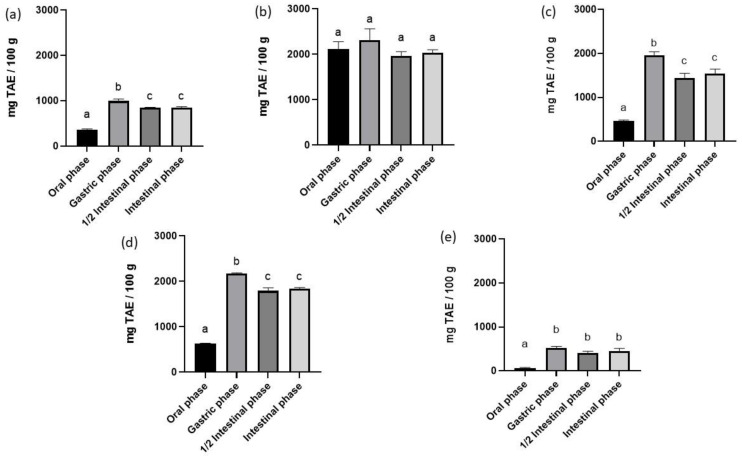
Total phenolic content (TPC) of in vitro digestion. TAE: Tannic acid equivalent. Data are presented as mean ± standard error of mean (SEM). (*n* = 3). Different superscript letters in columns indicate significant different data (*p* < 0.05). (**a**) hempseeds; (**b**) flowers; (**c**) hempseed protein extract; (**d**) soy protein extract; and (**e**) flaxseeds. End of oral phase, end of gastric phase, ½ intestinal phase, and end of intestinal phase.

**Figure 2 foods-12-00601-f002:**
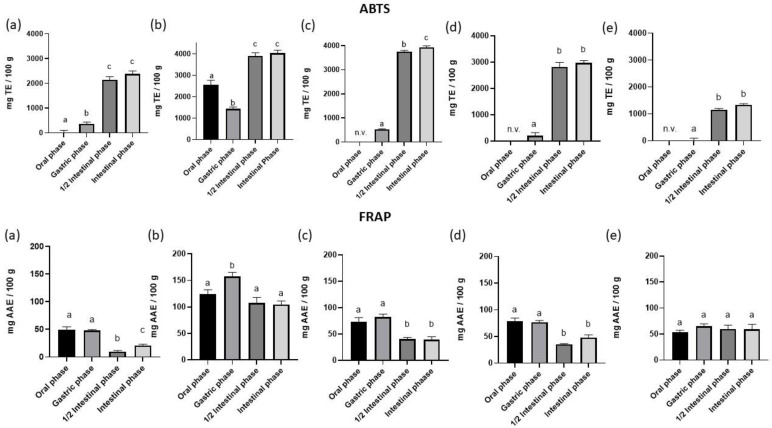
Antioxidant activity (ABTS and FRAP) of in vitro digestion. TE: trolox equivalent; AAE: ascorbic acid equivalent. Data are presented as mean ± standard error of mean (SEM). (*n* = 3). Different superscript letters in columns indicate significant different data (*p* < 0.05). (**a**) hempseeds; (**b**) flowers; (**c**) hempseed protein extract; (**d**) soy protein extract; and (**e**) flaxseeds. End of oral phase, end of gastric phase, ½ intestinal phase, and end of intestinal phase. n.v.: not valuable.

**Table 1 foods-12-00601-t001:** Chemical composition and digestibility of study matrices (% *w*/*w* on DM basis). Data are presented as mean ± standard error of mean (SEM). (*n* = 3). DM = dry matter; CP = crude protein; EE = ether extract; NDF = neutral detergent fibre; ADF = acid detergent fibre; ADL = acid detergent lignin. Different superscript letters in columns indicate statistically significant differences (*p* < 0.05).

SAMPLE	DM	CP	EE	NDF	ADF	ADL	ASHES	DIGESTIBILITY
Hempseeds	94.6 ± 0.12 ^a^	23.1 ± 0.57 ^a^	27.9 ± 0.75 ^a^	44.6 ± 0,21 ^a^	33.2 ± 0.31 ^a^	14.4 ± 0.32 ^a^	5.8 ± 0.10 ^a^	53.4 ± 0.78 ^a^
Flowers	98.4 ± 0.03 ^b^	18.9 ± 0.29 ^b^	12.9 ± 0.25 ^b^	37.0 ± 0,67 ^b^	23.7 ± 0.35 ^b^	11.7 ± 0.79 ^b^	16.3 ± 0.08 ^b^	37.4 ± 1.87 ^b^
Hempseed protein Extract	95.7 ± 0.02 ^c^	45.1 ± 0.84 ^c^	9.2 ± 0.09 ^c^	40.4 ± 1.80 ^a,b^	20.2 ± 1.02 ^c^	9.5 ± 0.62 ^b,d^	8.3 ± 0.08 ^c^	65.5 ± 0.67 ^c^
Soy protein extract	91.1 ± 0.22 ^d^	51.2 ± 0.57 ^d^	1.2 ± 0.06 ^d^	48.2 ± 0.97 ^a,c^	8.8 ± 0.14 ^d^	2.2 ± 0.10 ^c^	6.6 ± 0.20 ^d^	74.1 ± 3.59 ^c^
Flaxseeds	91.6 ± 0.14 ^d^	23.0 ± 0.45 ^a^	36.0 ± 0.07 ^e^	41.5 ± 1.27 ^a,b^	19.5 ± 0.35 ^c^	9.2 ± 0.41 ^d^	2.7 ± 0.20 ^e^	24.2 ± 2.08 ^d^

**Table 2 foods-12-00601-t002:** Total phenolic content (TPC) and antioxidant activity (ABTS and FRAP) of methanol extracts. TAE: tannic acid equivalent; TE: trolox equivalent; AAE: ascorbic acid equivalent. Data are presented ss mean ± standard error of mean (SEM). (*n* = 3). Different superscript letters in column indicate statistically significant differences (*p* < 0.05).

SAMPLE	TPC (mg TAE/100 g)	ABTS (mg TE/100 g)	FRAP (mg AAE/100 g)
Hempseeds	550.3 ± 28.27 ^a^	205.4 ± 3.37 ^a^	50.9 ± 4.30 ^a^
Flowers	2982.8 ± 167.78 ^b^	6122.1 ± 249.52 ^b^	123.6 ± 8.08 ^b^
Hempseed protein extract	568.9 ± 34.18 ^a^	174.5 ± 4.30 ^a^	29.73 ± 1.32 ^c^
Soy protein extract	792.3 ± 0.28 ^a^	209.6 ± 10.73 ^a^	17.4 ± 1.55 ^c,d^
Flaxseeds	634.0 ± 18.95 ^a^	73.2 ± 3.32 ^a^	10.4 ± 0.44 ^d^

## Data Availability

The datasets that were generated for this study are available on request to the corresponding author.

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
