# Peer review of "Total Phenolic Content and Antioxidant Activity of In Vitro Digested Hemp-Based Products"

_foods, 2023, doi:10.3390/foods12030601_

Round 1

Reviewer 1 Report

The title of the article is very confusing and it is not clear whether it is supposed to be a review article or original research. The phrase "for food and feed applications" is not suitable, because it is a general word and it is better to correct and improve the title of the article.

Abstract: The abstract should be more informative by giving real results rather than elastic sentences. Important and main contents should be given. Support the results with some quantitative data. Moreover, no conclusions are provided.

Abstract: line 14: statistics that need to mention the reference should not be included in the abstract, and in fact, the abstract should contain parts that the authors obtained from the results of their studies.

In the introduction, it is necessary to say something about this plant, its botany, its growth areas, and the research that have been done on it. In the last paragraph of the introduction, the purpose of the research must be clearly stated.

Introduction: what is the novelty of your work?

The paragraphs are too long. Check the paragraph extension in the manuscript.

It is necessary to use “2.1.materials” to express the materials used in the research.

At least the names of the analyzed compounds in part “2.1.Chemical Analysis” should be mentioned one by one.

2.2. Chemical Extraction: What material is intended to be extracted? Considering that the extraction of plant extracts in different solvents is considered, this title and related explanations must be corrected.

Because sections 2.4 are about the tests that are supposed to be performed on the extract, they should be listed under section 2.2.

Statistical Analysis should be numbered under the second section.

Why have you used the One-way Anova method for data analysis?

Discussion section: This part needs more specific detailed comparative studies. Please compare with similar works after presenting each result. Improve this section by this reference: 10.1186/s40538-022-00322-2

Conclusion: what is the future of your findings? The conclusion is not insightful, what are your suggestions?

Author Response

Dear Reviewer, thanks for your valuable comments. You will find the answers in the attached file. Thank you very much

Reviewer 2 Report

The article entitled “Nutritional and Functional evaluation of hemp-based products for food
and feed applications
is written well. The manuscript presentation is very good and need some minor corrections.

1. The keywords should be arranged alphabetically in the manuscript.

2. Unbold the references in braces i.e [1].

3. Figure 2: Improve resolution.

4.Figure 2: Unbold the bold words in title.

5. Figure 3: Improve resolution.

6. Line 115:  Replace & by “and”.

7.Table 1: The title of table should be written above the table not below the table.

8. Table 2: The title of table should be written above the table not below the table.

9. Line 278: The references should be written as [35-37].

10. Line 389: The references should be written as [43,51-53].

11.Line 305: write correct symbols of the ions i.e Fe+2 and Fe+3  

12. Why the standard deviations values are same in some experiments though experiments are different.

13. References: Reference no.16. The journal name should be abbreviated.

14. Conclusion: Revise the line 434-435.

15. Conclusions section: Conclusive remarks and further recommendation is necessary to be written in manuscript.

16. What is the practical potential of this research work?

17.The manuscript should be revised carefully. There are some minor grammatical mistakes.

18. Plagiarism should be checked.

Author Response

Dear Reviewer, thanks for your valuable comments. You can find the answers in the attached file. We hope it will be satisfactory. Thank you very much

Reviewer 3 Report

The research article “Nutritional and Functional evaluation of hemp-based products for food and feed applications” focuses on the characterisation of the nutritional and functional properties of hemp-based products, in particular total phenolic content (TPC) and antioxidant activities (FRAP, ABTS) of hempseeds (HS), hempseeds protein extracts, and flowers. The authors tested methanolic extract and samples obtained by in vitro digestion. Many papers focus on the same topic, so authors should emphasise what is a novelty in this paper. This paper need major revision. The authors should answer and resolve the following queries.

1. If the focus is on food/feed applications, why authors used methanol instead of ethanol or some other green solvent?

2. Why have they decided to use this method for digestion instead of the INFOGEST method? Why they did not use enzymes involved in polysaccharides (α-amylase, α-glycosidase) and lipid digestion (lipase)?

3. In Table 1, after the obtained values should be added statistical analysis (the letters are only in column DIGESTIBILITY, but the meaning was not explained after the table).

4. In lines 239 and 240 is written “The lipid and fiber content, which are higher for hemp, explain the lower digestibility of this matrix compared to soy”. However, it is not clear whether the authors compared hemp seed or hempseed protein extract with soy protein extract. If the letters in column DIGESTIBILITY show that there is a statistically significant difference between samples, then there is no statistically significant difference between hempseed protein extract and soy protein extract.

In lines 243 and 244 is written:” This finding is also confirmed by Wang et al., (2008) [29]. They not only observed that HS protein was highly digestible, but that the level of digestibility was also higher than that of soy.” This is the opposite of the previously written sentence. Please, check and better explain.

5. Why have the authors decided to follow only the change in total phenol content during in vitro digestion?

6. Explain if there is a possible reason that only in flowers there is not a difference in total phenol content between the oral and gastric phases.

7. Is there any correlation between TPC and antioxidant activity? In lines, 383 to 393 authors discuss the antioxidant activity of HS peptides. Is there a possibility to quantify their content in different phases of digestion?

Author Response

Dear reviewer. Thanks for your valuable comments. You will find the answers in the attached file. Thank you very much

Reviewer 4 Report

INTRO

- L.79-81: Authors talked only little about hemp used as food. Only one study cited (ref.9)? Please elaborate on how this research will fill the gap in demands for food and feeds. What this study did differently from ref.9?

M&M

- L.95-97: Please give details on specification of samples, sourcing, purity, etc.

R&D

- Table 1: Have authors tested significance for each measurement among samples? For example, for CP, does 51.2 (soy) differ from 23.1 (hempseeds)?

- L.238-240: Please support these statements with reference(s).

- L.245-248: Please support these statements with reference(s).

- Table 2: Please explain the variability in the Flowers’ TPC and ABTS dataset.

          Why mean values of Soy’s TPC was not different from Hempseeds and Hempseeds protein extract? The figures reported suggest the opposite. Same for mean value of Soy’s ABTS and Flaxseeds.

- L.260-263: Please explain this point based on Specification of raw material (previous comment for M&M, L.95-97)

- L.291-293: Same as previous comment. Further, do authors suggest using crude extract?

- L. 307-308: Could this observation be due to different mechanism (different targeting free radical) of methods used? Same to L.316-317.

- L.419-421: There is no basic information/ specification of materials used in this study (L.95-75).

- L.423-427: Since it is not clear whether it was a fare comparison between soy and hemp, this paragraph sounds like soy was demonized. Please support these statements with fact by such an environmental impact study as life cycle assessment or other sustainability index.    

Author Response

Dear reviewer, thanks for your valuable comments. You will find the answers in the attached file. Thank you.

Round 2

Reviewer 1 Report

The corrections were done.

Author Response

Dear Reviewer, thank you very much for your advice, which has enabled us to improve the quality of the article

Reviewer 3 Report

The authors  improved the manuscript. However, in my opinion, in the present form, this paper is not suitable for a high-ranked journal such as Foods.

Determination of total phenolic content (TPC) and antioxidant activities (FRAP, ABTS) are basic assays. 

Although often used, the method with Folin-Cicolateu reagent for determining the TPC has some limitations, since  FC reagent can also react with some other components (vitamin C, reducing sugars, sulfates). Quantification of the dominant polyphenolic compounds characteristic for this species and following the change in their content by HPLC analysis during in vitro digestion would be necessary. This part should be add.

In the manuscript, authors should explain why they have decided to use methanol instead of ethanol or some other green solvent. If the focus is on food and feed applications they should explain if and how methanol extracts could be used in industry. 

 In lines 223-226 is written “Zhou et al., (2020) [29] demonstrated that in vitro digestion of flaxseed polysaccharides did not lead to their degradation (no change in molecular weight), suggesting that ingested polysaccharides would reach the large intestine intact.” But Zhou et al. used human digestion model (“simulated saliva, gastric and small intestine conditions was assessed, as well as in vitro fermentation of FSP by human gut microbiota”), while authors used digestion model in pigs. Authors should explain how (to which extent) these models could be comparable.

In lines 245 -245 is written: “In our experiment data, HS protein extract and soy protein extract showed no statistically significant difference.” But it is not clear on which parameter/value authors think. 

In lines 366-371 is written: “In fact, in addition to being able to trap them, it reacts chemically through hydrophobic interactions, hydrogen bonding (oxygen atoms from polysaccharides' glycosidic chains and hydroxyl groups from phenolic compounds), and covalent bonds, thus reducing the bioavailability of phenolic compounds within the analysed matrices. Furthermore, as suggested by Ginsburg et al., (2012) saliva could play a key role in the solubilisation of phenolic compounds by substantially increasing their availability [54].”

 In the authors' opinion, according to obtained results, whether the bioavailability of phenolic compounds is increased or decreased after the oral phase?

Reference 22 is written in a different font.

Author Response

Dear reviewer, thanks for your advice, The answers to your  comments are attached, Thanks
